# Lightning declines over shipping lanes following regulation of fuel sulfur emissions

**Chris J Wright**[a], **Joel A Thornton**[a], **Lyatt Jaeglé**[a], **Yang Cao**[b], **Yannian Zhu**[b], **Jihu Liu**[b], **Randall Jones II**[a], **Robert H Holzworth**[c], **Daniel Rosenfeld**[d], **Robert Wood**[a], **Peter Blossey**[a], **and Daehyun Kim**[a,e]

[a]University of Washington, Department of Atmospheric Sciences, Seattle, WA, 98195
[b]Nanjing University, School of Atmospheric Sciences, Nanjing, China, 210023
[c]University of Washington, Department of Earth and Space Sciences, Seattle, WA, 98195
[d]The Hebrew University of Jerusalem, Institute of Earth Sciences, Jerusalem, Israel, 91904
[e]Seoul National University, Department of Atmospheric Science, Seoul, South Korea, 08820

**Correspondence:** Joel A Thornton (joelt@uw.edu)

**Abstract.** Aerosol interactions with clouds represent a significant uncertainty in our understanding of the Earth system. Deep convective clouds may respond to aerosol perturbations in several ways that have proven difficult to elucidate with observations. Here, we leverage the two busiest maritime shipping lanes in the world, which emit aerosol particles and their precursors into an otherwise relatively clean tropical marine boundary layer, to make headway on the influence of aerosol on deep convective clouds. The recent seven-fold change in allowable fuel sulfur by the International Maritime Organization allows us to test the sensitivity of the lightning to changes in ship plume aerosol number-size distributions. We find that, across a range of atmospheric thermodynamic conditions, the previously documented enhancement of lightning over the shipping lanes has fallen by over 40%. The enhancement is therefore at least partially aerosol-mediated, a conclusion that is supported by observations of droplet number at cloud base, which show a similar decline over the shipping lane. These results have fundamental implications for our understanding of aerosol-cloud interactions, suggesting that deep convective clouds are impacted by the aerosol number distribution in the remote marine environment.

## Introduction

By acting as cloud condensation nuclei (CCN), aerosol particles influence clouds and, in turn, the Earth's energy balance. Aerosol-cloud interactions represent a significant uncertainty in our understanding of the Earth's climate (Boucher et al., 2013). Maritime ship traffic leads to the emission of aerosol particles and associated precursors into relatively clean marine air. These emissions enable study of how increased CCN perturb low-level marine stratus cloud droplet number distributions and related cloud macrophysical properties, such as cloud albedo and lifetime.(Diamond et al., 2020; Yuan et al., 2022; Durkee et al., 2000; Radke et al., 1989).

Deep convective cloud (DCC) systems occur throughout the tropics, and are essential to the Earth's water and energy cycles (Feng et al., 2021). However, there is little consensus on the mechanisms or magnitudes of aerosol particle impacts on DCCs (Tao et al., 2012; Seinfeld et al., 2016; Igel and van den Heever, 2021; Varble et al., 2023; Williams et al., 2002). Thornton et al. (2017) documented a potential case of persistent maritime aerosol-DCC interactions analogous to stratocumulus ship tracks, with the discovery of enhancements in lightning over major shipping lanes passing through the Indian Ocean and South China Sea (Fig. 1).

Several mechanisms have been proposed to explain how aerosol particles from ship emissions could enhance lightning frequency, all of which involve enhanced cloud droplet nucleation (Twomey, 1977), leading to either 1) perturbations to super-cooled liquid water and ice hydrometeor distributions and enhanced charge separation in the mixed-phase region of DCC (Mansell and Ziegler, 2013; Blossey et al., 2018; Sun et al., 2024; Takahashi and Miyawaki, 2002; Deierling et al., 2008); or 2) an increase in the frequency or intensity of deep convection due to changes in the vertical distribution of humidity (Abbott and Cronin, 2021) or heating (Fan et al., 2018; Grabowski and Morrison, 2020). Some combination of 1 or 2 is also possible.

In January 2020, the International Maritime Organization (IMO) reduced the amount of allowable sulfur in fuel by a factor of seven, from 3.5% to 0.5% to curb effects of maritime shipping on air pollution (IMO, 2020). Recent analyses of shallow stratocumulus marine clouds over shipping lanes find changes to cloud brightness, droplet number, and droplet size associated with the IMO regulation, presumably due to the shift in aerosol number-size distribution (Watson-Parris et al., 2022; Yuan et al., 2022; Diamond, 2023).

We investigate whether the IMO fuel sulfur regulation has impacted lightning over the shipping lanes in the tropical Indian Ocean and South China Sea. We find the shipping lane lightning enhancement decreases significantly with the onset of the IMO regulation and that this decrease persists across a range of atmospheric conditions. We further show that the mean cloud droplet number concentration of shallow warm clouds over the Indian Ocean shipping lane was enhanced before the IMO regulation and also exhibits a decrease since the IMO regulation. We discuss the implications of these new results for mechanisms of shipping lane lightning enhancement and aerosol-DCC interactions.

## Approach and findings

The Port of Singapore accounts for 20% of the world's bunkering fuel demand. The two primary shipping lanes it services—the Indian Ocean and South China Sea (hereafter "the shipping lanes")—have nearly an order of magnitude higher traffic than other shipping lanes around the world (Figure 1, top panel) (Mao et al., 2022). As shown in Figure 1 (middle panel), prior to 2020, the mean absolute lightning stroke density measured by the World Wide Lightning Detection Network (WWLLN) remains enhanced over these shipping lanes, consistent with Thornton et al. (2017). Since 2020, however, when the IMO regulation of sulfur emissions began, lightning over the shipping lanes has decreased to an annual stroke density about 1 stroke $km^{-2}$ $year^{-1}$ lower than before the regulation (Figure 1, bottom panel). While some of the largest absolute declines in lightning since 2020 occur over the shipping lanes, lightning has increased or decreased in other parts of this region as well. As we illustrate below, variability in the dynamic and thermodynamic context for convection over these shipping lanes must be taken into account to better isolate the potential impacts of shipping emissions.

Regional ship traffic, as measured by vessel fuel sales at the Port of Singapore, has been relatively constant or even increased since 2020 (Figure S1) (Port of Singapore, 2024). The disruption by COVID-19 did not obviously decrease activity at the port, seeming only to have briefly slowed the growth of cargo throughput for 2-3 months in 2020 (Gu et al., 2023). Therefore, we focus on controlling for the variability in background meteorological conditions that impact the frequency and intensity of convection, and thus lightning, over the shipping lanes.

We first examine the shipping lane lightning enhancement using two controls on background meteorology: 1) we only sample precipitating clouds(Huffman et al., 2015; Pradhan and Markonis, 2023; Watters et al., 2023); and 2) we restrict analyses to the specific seasons in each region favorable for lighting (November to April in the Indian Ocean; June to November in the South China Sea). Using these criteria, we composite lightning observations as a function of distance to the shipping lanes, the center of which we define as the peak in shipping emissions from the EDGAR emissions inventory (see Methods). As shown in Figure 2a, mean absolute lightning exhibits a clear enhancement over the shipping lane before 2020 (pre-IMO), between approximately 150km south to 150km north of the shipping lanes, and that has decreased since the regulation onset in 2020 (post-IMO) (Figure 2a).

To account for inter-annual variability in the frequency and intensity of convection in the region, we regress the observed annual lightning at a given distance from the shipping lane against three variables known to relate to lightning frequency (Convective Available Potential Energy (CAPE, discussed further below), precipitation rates (Romps et al., 2018), and the annual mean Oceanic Niño Index (ONI)) as well as several spatial variables such as latitude and longitude (Appendix A). Inter-annual variability in the MJO was small and had a negligible impact when included in the regression (see SI). The regressed variables explain 65% of variance of the annual means. We subtract the regressed lightning

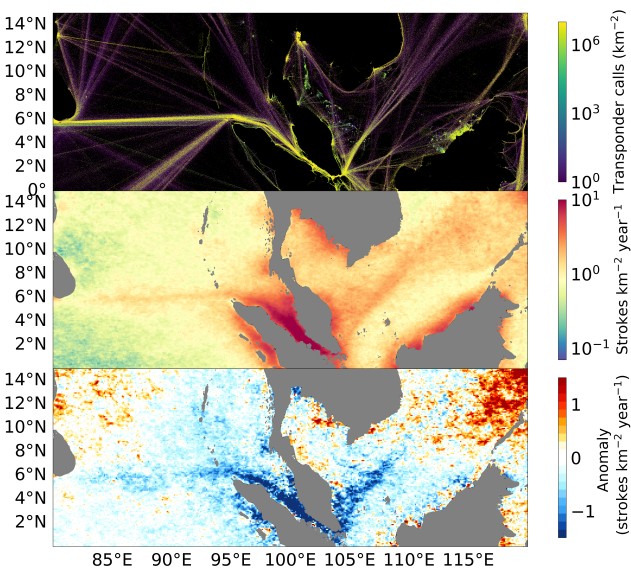

**Fig. 1.** (top) Map showing the total number of Automatic Indentification Systems (AIS) transponder calls from 2015-2021, used by maritime vessels for collision avoidance. Data from the IMF World Seaborne Trade dataset (Cerdeiro et al., 2020). (middle) Climatological mean lightning stroke density near the Port of Singapore (2010-2019). (bottom) Difference of the post-regulation period (2020-2023) lighting stroke density from the 2010-2019 climatology above. See Appendix for further discussion of the retrievals.

from the observed annual mean, leaving the anomalous mean lightning stroke density that cannot be explained by interannual variability in storm occurrence and intensity, shown in Figure 2b.

The annual anomalous enhancement in lightning over the shipping lanes prior to 2020 is even clearer after regressing out meteorological variability, as is the near step-change decrease in the anomaly after 2020 (2b). Prior to the IMO regulation, essentially 100% of fuel sold at the port was high-sulfur (2b, right axis); correspondingly, the lightning anomaly over the shipping lane was 3.9 strokes $km^{-2}$ $year^{-1}$ on average and was never below 2.5 strokes $km^{-2}$ $year^{-1}$ for more than one year at a time. Adoption of the IMO regulation was prompt in 2020, as indicated by the change in high-sulfur fuel from 100% to less than 35% of fuel sold at the Port of Singapore. The Port of Singapore experienced little attenuation of fuel sales at the onset of COVID-19 (Gu et al., 2023), and total fuel sales have increased since 2020 consistent with higher traffic (Figure S1). Since 2020, the shipping lane lightning enhancements compared to adjacent regions have declined by ~67% to 1.25 strokes $km^{-2}$ $year^{-1}$ on average.

To further control for higher-frequency variations in convective activity and intensity, we examine the lightning enhancement in a 2-dimensional CAPE and precipitation space, using 3-hourly coincident observations of CAPE, precipitation, and lightning. Cheng et al. (2021), building on Romps et al. (2018), showed that CAPE×precipitation is a reasonable proxy for tropical oceanic lightning frequency, given that a CAPE threshold is implemented. The 3-hourly CAPE and precipitation observations implicitly capture variability arising from more indirect sources, such as sea surface temperatures (SST), MJO events, fronts, etc (see SI for further discussion).

We compute lightning frequency in each CAPE-Precipitation bin using data from a region centered over each shipping lane and from reference regions adjacent to the shipping lanes (see Figure S2). We then compute a relative enhancement in lightning over the shipping lanes, before and after the onset of the IMO regulation, by taking the difference between corresponding CAPE-Precipitation bin-means in the shipping lane and associated reference box. The resulting shipping lane lightning enhancements as a function of both CAPE and Precipitation are shown in Figure 3. Before the IMO regulation (Pre-IMO), a shipping lane lightning enhancement existed in nearly every thermodynamic setting (e.g.,

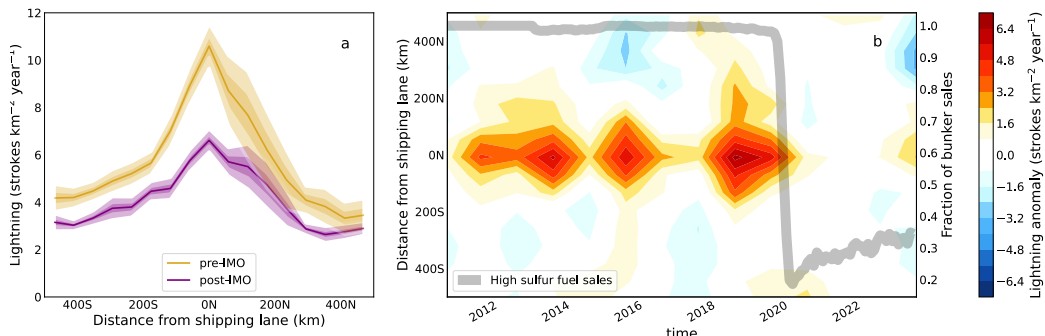

**Fig. 2.** (a) Lightning stroke density composited as a function of distance to the shipping lanes before and after the IMO regulation. Shading represents ±2SE and ±3SE (b) Hovmöller diagram of the annual mean lightning anomaly from the linear regression using Convective Available Potential Energy, precipitation, and Oceanic Niño Index from reanalysis data and observations, as well as spatial variables (latitude, longitude, lat*lon, lat², lon²)(see text and SI for more details). Gray line shows the fraction of fuel sales that are high sulfur fuel at the Port of Singapore.

in each CAPE-Precipitation bin, Figure 3a,d) for both shipping lanes. As indicated by the larger pre-IMO perturbation, it seems that low-CAPE environments may have been more susceptible to aerosol enhancement of lightning.

Since the IMO regulation (Post-IMO), both shipping lanes exhibit significantly weaker lightning enhancements across most CAPE-Precipitation regimes (Figure 3b,e). The bin-by-bin differences between the Pre and Post-IMO lightning enhancement histograms are shown in Figure 3 (c and f). On average across all CAPE-Precip conditions, the lightning enhancement has decreased by 76% and by 47% for the Indian Ocean and South China Sea shipping lanes, respectively (Figure 3c,f). That is, for the same convective setting characterized by CAPE and Precipitation rates, the enhancement in lightning over the shipping lanes (as compared to adjacent regions) is significantly smaller after 2020 than it was before 2020.

Based upon the above, we hypothesize that the decline in the lightning enhancement since 2020 is most consistent with the IMO regulation changing CCN in the region. If decreasing sulfur emissions over the shipping lanes has reduced the total number of viable CCN and disrupted an associated mechanism for lightning enhancement, then there should be a corresponding change in warm cloud microphysics. To further test our hypothesis, we use Moderate Resolution Imaging Spectroradiometer (MODIS) satellite observations of cloud droplet number ($N_d$) in low clouds over the Indian Ocean shipping lane, where the influence of land is weaker and ship emissions are stronger, during the high-lightning season. The retrievals of $N_d$ follow the method outlined in Zhu et al. (2018)(see Appendix A). Retrievals of $N_d$ can only be done for shallow cumulus, not DCC. As a result, $N_d$ retrievals sample a different set of conditions than the lightning observations. We assume that the behavior of $N_d$ in shallow cumulus clouds from the same region is related to, though not necessarily a direct proxy for, $N_d$ at cloud base in DCC.

In Figure 4, we show that prior to the IMO regulation, there is a clear trend in $N_d$ toward land (north), as well as a clear perturbation in $N_d$ over the shipping lane. This $N_d$ perturbation is roughly 10-15% above the average of droplet concentrations 150km north and 150km south, which is larger than the shipping lane perturbations to $N_d$ detected by Diamond et al. (2020) in Southeast Atlantic stratoculumus clouds. The $N_d$ perturbation over the Indian Ocean shipping lane is a significant finding on its own, as observations of persistent, mean-state $N_d$ perturbations by ships are rare (Diamond and Wood, 2020), especially for convectively active regions we show here.

Since the IMO regulation, the $N_d$ away from the shipping lane mostly maintain their previous levels, as indicated by the overlap in the 95% confidence intervals, particularly to the north (upwind). Meanwhile, the enhancement in $N_d$ over the shipping lane has become essentially undetectable 4. The decline in $N_d$ over the shipping lane relative to the surrounding region establishes additional support for a relationship between the declining lightning enhancement and a shift in aerosol particle number-size distributions over the shipping lanes induced by the IMO regulation. $N_d$ derived from shallow cumulus clouds will not be directly proportional to the CCN available for activation

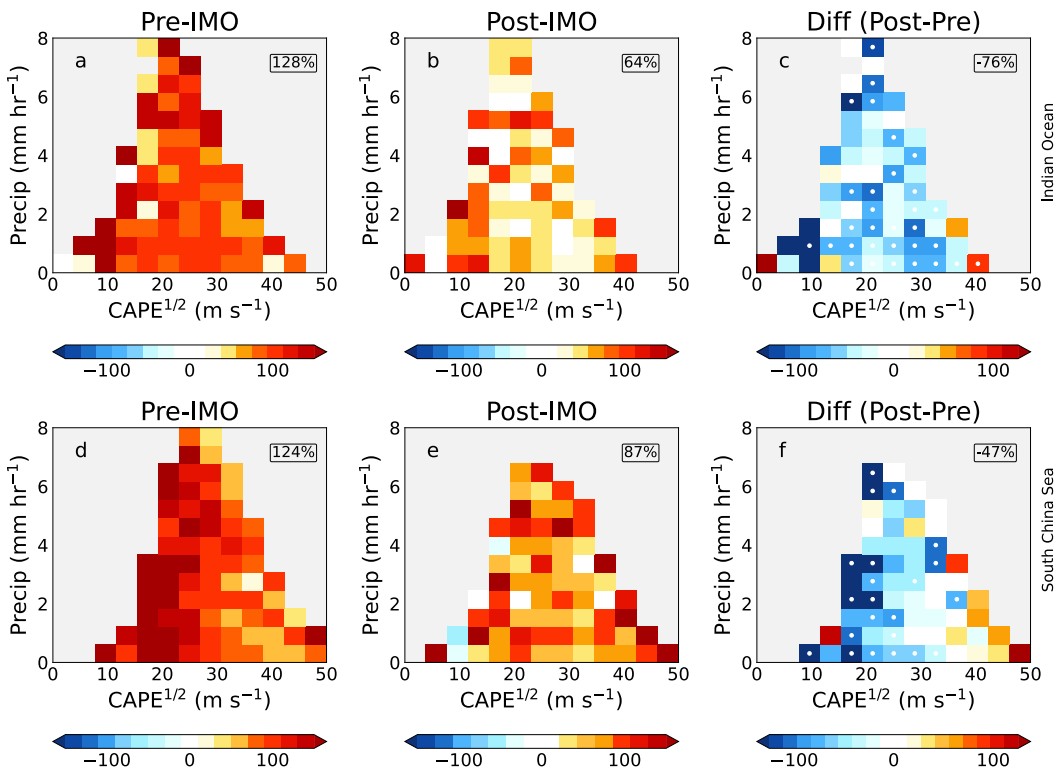

**Fig. 3.** Mean shipping lane percent enhancement in lightning stroke density (i.e., the relative difference in lighting over the shipping lane from that over immediately adjacent regions, see text), shown as colored pixels, binned by square root of CAPE reanalysis data (x-axis) and precipitation observations (y-axis) for the Indian Ocean (a)–(c) and South China Sea (d)–(f) shipping lanes. Enhancements since the regulation (b, e) are lower than before the regulation (a, d). The difference between post- and pre-IMO periods of the shipping lane lightning enhancements are represented in (c, f), where stippled bins indicate significance (p less than 0.05).

in high-supersaturation DCC (Hobbs et al., 2000), nor to the lightning enhancements that might result from CCN enhancements. However, the
change in $N_d$ is an additional observable indication, independent of the lightning observations, that the IMO regulations have clearly shifted
aerosol particle distributions over the shipping lanes of interest here.

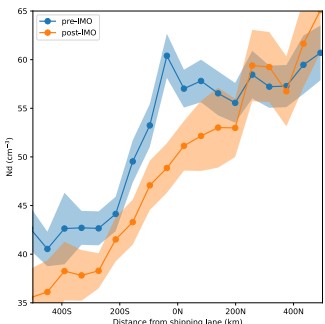

**Fig. 4.** Warm cloud-base droplet number ($N_d$) concentrations over the Indian Ocean, derived from MODIS observations of optical depth and effective radius following the procedure from Zhu et al. (2018). The pre-IMO regulation period is 2010-2019. Note the general increasing trend northward through the domain toward greater land influence, and the broad localized enhancement over the shipping lane prior to the regulation. To remove the effects and trends of large biomass burning and dust events, we remove any $N_d$ retrievals where dust concentrations increase above 1 µg m−3 or black carbon concentrations above 0.1 µg m−3 (approximately the 50th percentile in each case). Shading represents ±2SE.

## Implications

We find that a previously identified enhancement in lightning stroke density over the two major shipping routes near the Port of Singapore has
declined by over 40% since 2020, when the IMO regulation of maritime shipping sulfur emissions came into effect. The decline is evident after
controlling for natural variations in environmental conditions that characterize the convection intensity and frequency (see Figure 3). While
ships may act as lightning attractors (Peterson, 2023), there is not evidence of a change in the number of ships traversing these shipping lanes
over this time period (Figure S1). Further, an independently observed perturbation to $N_d$ over the Indian Ocean shipping lane prior to the IMO
regulation has since nearly vanished, indicating a coincident change in CCN over the region. The concomitant decline of $N_d$ and lightning,
timed with the onset of compliance with the IMO regulation, provides new evidence to support the set of CCN-mediated hypotheses previously
proposed for invigoration of lightning (Fan et al., 2018; Abbott and Cronin, 2021; Mansell and Ziegler, 2013; Grabowski and Morrison, 2020).
Precisely how increases in CCN and $N_d$ caused by ship emissions might lead to enhanced lightning remains unresolved. Both the pre-IMO
and post-IMO perturbations to $N_d$ are smaller than the enhancements of lightning, which may be explained by high scavenging rates in DCC
during the high lightning season, as well as the different supersaturation conditions sampled by the $N_d$ (shallow cumulus) and lightning
retrievals (deep convection), non-linear relationships between $N_d$ and ice secondary ice production, and ice nucleation.
Seppälä et al. (2021) find that a factor of 10 reduction in ship fuel sulfur content shifts emitted aerosol particles to smaller sizes and lower
total number concentrations. Ultrafine particles <50nm increase and those >50nm decrease substantially, which implies that ultrafine particle
invigoration of updraft velocities proposed in Fan et al. (2018) was not contributing to the shipping lane lightning enhancement pre–IMO. We
conclude the lightning enhancement prior to the IMO regulation was mostly the result of higher concentrations of larger aerosol particles
(e.g. >50 nm) that perturbed: 1) cloud microphysics, such as elevated supercooled liquid water concentrations or rime splintering (see SI)
(Mansell and Ziegler, 2013); and/or 2) updraft frequency, by means of heightened free tropospheric humidity or a mesoscale circulation
response (Blossey et al., 2018; Grabowski and Morrison, 2020). Enhancements in ultrafine particles may play a role in the smaller non-zero
shipping lane lightning enhancement that has persisted post-IMO.
The IMO regulation of ship fuel sulfur illustrates connections between international trade, air pollution, DCC microphysics, and lightning.
Further work combining in situ and remote sensing of aerosol and cloud microphysics together with lightning frequency is needed to clarify the
mechanisms behind these connections, and to quantify the relative roles of dynamic and microphysical responses. Our findings herein show
that these regions remain a useful testbed for understanding aerosol pollution impacts on DCC and lightning.
**Appendix A: Retrievals, data processing, and methods**
Lightning stroke density observations come from the Worldwide Lightning Location Network (WWLLN), a ground-based lightning detection
network with continuous global coverage of lightning at a resolution of 10km (Dowden et al., 2002). WWLLN uses very low frequency radio
impulses (3-30 kHz) that, upon emission from a lightning stroke, propagates between the Earth-ionosphere waveguide and disperses into a wave
train. The phase and frequency of that wave train determine the time of group arrival at three or more measurement stations, which can be used
to back out the location of the stroke. While the detection efficiency for individual events is lower than satellite-based methods, continuous
observations for more than a decade offer much more statistical power over our region of interest.
We use integrated multi-satellite retrievals for GPM (IMERG) precipitation rates (Huffman et al., 2015) and European Centre for
Medium-Range Weather Forecasts (ECMWF) ReAnalysis-5th Generation (ERA5) atmospheric reanalyses (Hersbach et al., 2020) CAPE to
compare the enhancement across various thermodynamic conditions. IMERG precipitation combines microwave and radar retrievals from
TRMM and the GPM constellation. In ERA5, a value for CAPE is calculated for every departing level between the surface and 350hPa as
follows:

$$CAPE = \int_{z_{dep}}^{z_{top}} g \left( \frac{\theta_{ep} - \overline{\theta}_{esat}}{\overline{\theta}_{esat}} \right) dz$$

where $z_{dep}$ is the departing level, $z_{top}$ is the level of neutral buoyancy, $\theta_{ep}$ is the virtual potential temperature of the parcel, and $\overline{\theta}_{esat}$ is the
saturation virtual potential temperature of the environment. Once CAPE has been calculated for all levels, the most unstable layer is selected.
We use CAPE$^{1/2}$, which is directly proportional to w$_{max}$, the theoretical maximum vertical velocity achievable at a location given the stability
of the atmosphere. This follows from the proportionality between kinetic energy and the square of velocity. Further discussion of CAPE as it
relates to lightning can be found in (Cheng et al., 2021) and (Romps et al., 2018).
Lightning in Figure 1 is shown on 0.1º x 0.1º grid, calculated from 3-hourly lightning stroke densities. For subsequent calculations of
the enhancement (Figures 2-3) all data (CAPE, precipitation, and lightning) is 3-hourly and mapped to a 0.5ºN x 0.625ºE grid to minimize
collocation errors and noise, and for comparison with MERRA-2 aerosol and meteorological reanalysis fields. Smoothly varying data (CAPE)
is remapped bilinearly, while non-smoothly varying data (precipitation and lightning) are remapped conservatively (see Staff (2014) and sources
therein for further detail on regridding practices). To provide some basic control for thermodynamic and meteorological variability, we only
consider precipitating clouds (precipitation greater than 0.1mm/hr) during the high-lightning season (see SI).
We use data from 2010 onward, as WWLLN detection efficiency was still increasing rapidly prior to 2010. The shipping lanes are defined as
regions where the Emissions Database for Global Atmospheric Research (EDGAR) PM$_{2.5}$ shipping emissions are greater than $5\times10^{-12}$ kg m$^{-2}$
s$^{-1}$ (Crippa et al., 2016). To remove influence from katabatic flows and sea-breeze driven convergence, we only consider the larger blue regions
outlined in Figure S2. This notably removes the straight of Malacca, a region with both very high shipping emissions and active convection.
There, surface convergence from land-based precipitation outflows on Sumatra and Malaysia and the adjacent landmasses make it challenging
to establish a counterfactual, given the well-known land-ocean contrast in lightning stroke rates Cheng et al. (2021); Romps et al. (2018).

For Figure 2, the lightning stroke density (F) as a function of time (t) and distance from shipping lane (y) from the entire record is regressed as:

$$F(y,t) = \beta * X(y,t) + \epsilon$$

where X is the vector of predictors, (CAPE, precipitation, ONI, latitude (lat), longitude (lon), lat*lon, $lat^2$, $lon^2$), $\beta$ is the vector of coefficients. $\epsilon$ is the residual or "anomaly" that is shown in the figure. This anomaly represents the difference between the lightning one would expect given the environmental conditions ($\beta * X$) (see Figure S3) and the observed lightning ($F$). In accordance with Cheng et al. (2021) CAPE has been set to zero where $CAPE^{1/2}$ ¡ 15 ms$^{-1}$.

We utilized the "brightest 10%" method (Zhu et al., 2018; Cao et al., 2023) to obtain reliable $N_d$ cloud droplet number concentration retrievals from MODIS Aqua across our target domain from 2010 to 2023. This method involves selecting the brightest 10% of clouds within each scene to calculate $N_d$ values for every 0.5° x 0.5° grid box. The validity of this retrieval method has been corroborated through comparisons with ship-based observations (Efraim et al., 2020; Wang et al., 2021). $N_d$ is computed using the cloud effective radius ($r_e$) and cloud optical depth ($\tau$), as described by the equation:

$$N_d = \frac{\sqrt{5}}{2\pi k} \left( \frac{f_{ad} \, C_w \, \tau}{Q_{ext} \, \rho_w \, r_e^5} \right)^{\frac{1}{2}}$$

where $k$ represents the volume radius ratio of cloud droplets ($r_v$) to re ($k = (r_v/r_e)^3 = 0.8$). The term $f_{ad}$ denotes the adiabatic fraction, for which we assumed a constant value of 1 in our study, due to the absence of more refined alternatives (Bennartz and Rausch, 2017; Grosvenor et al., 2018). $C_w$ signifies the adiabatic cloud water condensation rate within an ascending cloud parcel, expressed in grams per cubic meter per meter (g m$^{-3}$ m$^{-1}$). The extinction efficiency factor, $Q_{ext}$, is assumed to be 2, and $\rho_w$ is the density of water. To enhance the accuracy of our $N_d$ estimations for each 0.5° x 0.5° grid box, we excluded pixels where the solar zenith angle exceeded 65 degrees (Grosvenor and Wood, 2014). We also excluded of scenes containing mixed-phase, ice, or multilayer clouds. Consequently, after applying these filtering criteria, the remaining dataset comprised less than 1% of multilayer cloud pixels in any given grid. We use only the Indian Ocean shipping lane to maximize signal-to-noise, as the South China Sea has a much weaker signal due to its proximity to land and lower ship emissions. Inclusion of the South China Sea in the analysis does not alter the results. Finally, in order to remove the impact of dust storms advected over the Bay of Bengal, and to thereby reduce interannual variability in $N_d$ outside the shipping lanes, we collocate 3-hourly MERRA-2 aerosol reanalysis output of dust and black carbon with the MODIS $N_d$ retrievals. We then remove any $N_d$ retrievals where dust concentrations increase above 1 ng m−3 or black carbon concentrations above 0.1 ng m−3 (approximately the 50th percentile in each case). Limited observations in the region likely hinder the ability of reanalysis products to capture the full variability in CCN sources (see SI for discussion of aerosol optical depth), possibly explaining some remaining differences in $N_d$ over the southern region of the domain pre and post 2020.

**Data availability**

ERA5 CAPE may be downloaded using the Copernicus API at cds.climate.copernicus.eu. IMERG Precipitation and MERRA-2 aerosol are available for download at disc.gsfc.nasa.gov. ONI index is available at psl.noaa.gov/data/correlation/oni.data. Precipitation Feature reflectivity datasets are available for download at: https://atmos.tamucc.edu/trmm/data/. MODIS Aqua (MYD06) retrievals are available at ladsweb.modaps.eosdis.nasa.gov. Precipitation Feature reflectivity datasets are available for download at: https://atmos.tamucc.edu/trmm/data/. Global ship traffic density is available at: datacatalog.worldbank.org/search/dataset/0037580/Global-Shipping-Traffic-Density. Analysis and plotting available at 10.5281/zenodo.11373991 (Wright, 2024). WWLLN lightning location data are collected by a global scientific

collaboration and managed by the University of Washington. The WWLLN collaboration receives no federal, state or private funds to pay for the network operations, which are fully paid for by data sales (available at https://wwlln.net). Therefore, the stroke-level data is not free to the public. The composited annual stroke densities (as a function of distance from the shipping lane) and the mean pre- and post-regulation stroke densities region-wide are provided as part of the Zenodo code supplement.

**Author Contributions**

Analysis: CJW. Writing: CJW and JAT. Conceptualization and methodological development: CJW, JAT, LJ, and RW. $N_d$ retrievals by: YC, YZ, JL. Additional expertise provided by RH, DR, RJ, PN, and DK

**Acknowledgements**

This work was funded by a grant from the U.S. National Science Foundation (AGS-2113494). Additional funding included (in order of authorship): Natural Science Foundation of China grant 42075093 (YC, YZ, JL), BSF Grant 2020809 (DR), NASA/UMBC grant NASA0144-01 (RW), NSF grant AGS-1912130 (PN), and New Faculty Startup Fund from Seoul National University (DK). The authors wish to thank the World Wide Lightning Location Network (http://wwlln.net), a collaboration among over 50 universities and institutions, for providing the lightning location data used in this paper.

**Competing interests**

The contact author has declared that none of the authors has any competing interests.

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
