# Peer review of "Lightning declines over shipping lanes following regulation of fuel sulfur emissions"

_EGUsphere, 2024_

## Referee Comment (RC1)

**General comments**:

Wright et al. analyze the response of a previously noted enhancement in lightning to the IMO 2020 sulfur fuel regulation, finding that the enhancement is reduced following a shift toward low sulfur fuels, and that this change is likely caused by decreased CCN activity. The manuscript presents a thorough analysis of many factors that correlated with lightning activity, including CAPE, precipitation, and ENSO, and additionally confirming a likely enhancement in cloud-base number concentration based on warm clouds. A few sections in the manuscript would benefit from rewording and/or clarification. In addition, the WWLLN data used in their analyses are not freely available to the public (but rather must be purchased). Aside from these concerns, this article presents a thorough, concise, and important set of findings and commentary.

**Specific comments:**

- The abstract claims to test the sensitivity of lightning to "aerosol size distributions," yet the paper does not quantitatively present any results regarding aerosol sizes before and after IMO-2020. A better descriptor would be "aerosol concentration" or "aerosol emissions."
- Figure 2 and associated discussion: does this analysis include both the Indian Ocean and South China Sea composited, or only the Indian Ocean? Please clearly state.
- Page 3, 2D analysis of 3h CAPE / precip space:
  - What percent variance in lightning can be captured on the 3-hourly timescale, compared to the annual regression discussed earlier? Other works have indicated that CAPE and precipitation are not the best markers of convective strength over the ocean (see e.g. M.R. Igel 2014), so it would be beneficial to provide a quantification (and potentially brief discussion) of the predictive relevance of CAPE and precip in these data, rather than relying on Cheng 2021.
  - If following Cheng 2021, Fig. 2 should exclude points where $CAPE^{(1/2)} < 15$ m/s.
  - It is potentially interesting that lower-CAPE retrievals show both a stronger pre-IMO enhancement, and a greater difference following IMO, particularly in the South China Sea. This would indicate that weaker systems (lower CAPE) are more susceptible to aerosol, which may warrant a brief discussion.
- Page 4, discussion of optical thickness, reads "We have partially accounted for....using MERRA-2 reanalysis estimates...in constructing Figure 4." However, it is

not clear from the figure caption how this correction is performed. Presumably the phrases at the end of the Appendix explain this correction, and should be referenced in the text accordingly.

- The Supporting Information would benefit from subheadings to organize and divide contents. References to the SI in the main text would then be more precise.
- SI figure 3:
  - Clarify whether the data displayed are 3-hourly or annual mean
  - SI page 2 indicates that S3 shows data "outside of the shipping lane", but the figure appears to include all data, including over top of the shipping lane.
- ACP Data Availability policy requires that data which cannot be deposited publicly because of commercial constraints should include a detailed explanation of why this is the case, and additionally that the data should be made available to reviewers. The existing statement in the manuscript only directs the reader to WWLLN.net, where data are only accessible for a fee, and should be updated to reflect ACP's requirements.

**Technical comments:**

- I suggest the authors confirm that the manuscript falls within ACP's 2500 word limit – a cursory word count on my part read 2700, but this included in-text citations which may not count toward the limit.
- I did not immediately notice any typographical errors and applaud the authors on their writing.

---

## Author Comment (AC1)

**RE: ACP MS # 2024-3236 Response to Reviewers Comments**

We thank the editor for the opportunity to provide responses to the reviewers' concerns with our recently submitted manuscript. We also thank the reviewers for their thoughtful assessment of the manuscript and for providing useful perspectives and suggestions that will undoubtedly improve the communication of our findings.

**Point-by-point response**

**Reviewer 2**

*Main points*

"While there has been a decrease in lightning in the shipping lanes, there has also been a broad decrease in lightning across the study region (but with increases in some regions, Fig 1c). Given the pattern of lightning in the study region includes the shipping lanes, a large-scale reduction in lightning frequency across the region would also show a larger reduction in the shipping lanes. How do we know that the observed shipping lane change is not part of a larger change in lightning/convective cloud occurrence across the region?"

We address this concern with the comments below, along with our analyses in updated Figures 2 and 3. Figure 1 is intended to give the reader a view of what is, more or less, the "raw" data, without controls.

"It is not clear to me that the meteorological parameters used for regressing out the background meteorological state work well. The manuscript states that they only explain 33% of the variance in lightning occurrence."

We have updated Figure 2 to include a linear model with improved explanatory power. This is done by adding spatial variables to the regression, including latitude, longitude, latitude*longitude, $lat^2$, and $lon^2$ (see e.g., Diamond et al, 2020) The **new explained variance is 65% ($r^2 = 0.65$)**.

"Fig. 3 suggests that they don't have a strong impact to lightning frequency (or enhancement), that there is a change in lightning occurrence across all meteorological conditions. I find this a little concerning, as we might expect the aerosol enhancement of convective clouds to depend on cloud/meteorological regime - why does it not in this case? This could be a potential indicator of some kind of confounding effect."

There is indeed a strong impact of CAPE and precipitation on lightning *frequency* (see e.g. Cheng et al, 2021). Over the shipping lane, the relationship looks quite similar:

[Figure]

*Figure 1. Lightning stroke frequency in CAPE-Precip space, pre-IMO regulation in the Bay of Bengal.*

As noted by Reviewer 1, the enhancement in Figure 3 is not necessarily completely uniform in magnitude — the South China Sea, in particular, shows stronger enhancements and declines in low-CAPE regimes. Still, we agree that there is not strong variability in the enhancement across CAPE–Precip space, perhaps because **these variables are generally indicators of lightning generation potential, not of cloud susceptibility to aerosol**. Factors that do represent cloud susceptibility to aerosol, i.e. boundary layer coupling and, more generally, the convective lifecycle, are not apt to be represented by the CAPE-Precip space (nor are they intended to be). Aerosol impacts over the shipping lane throughout various coupling states and the convective lifecycle is the subject of ongoing study by the co-authors.

*Minor points*

"Abstract (and elsewhere) - while there have been changes to the aerosol size distribution, is the main effect a change to the number concentration? I might have expected this to be the first-order effect?"

Our intention with "size distribution" was to elicit thoughts of both 1) direct emission of CCN from ships and 2) growth of aerosol to CCN-size via deposition of oxidized sulfur emissions. We will instead use size-number distribution to avoid confusion.

"Is a shallow-cloud Nd product suitable here? The bright core adiabatic assumption may be suitable for non-precipitating stratocumulus, but in more convective cases, it seems that it will pick out the precipitating locations (which are not adiabatic). Comparisons with aircraft data suggest it is not reliable in convective cases (Gryspeerdt et al, ACP, 2023)."

The core represents the freshest convection and, therefore, the strongest connection with underlying CCN. For this reason, we prefer to use the bright core. Over this region, a majority of clouds are precipitating. The accuracy of the Nd of the cores is lower because the cloud optical depth is larger and therefore the relative accuracy becomes smaller, but not necessarily biased; e.g., the susceptibility of cloud properties to Nd is not affected by the method of its retrieval, core or average area (Wang et al, 2023).

"P1P3L3 - Williams et al, JGR 2002 might also be a relevant paper here

Thank you for your comment. The citation has been added to paragraph two, which discusses the lack of consensus on aerosol-convection-lightning interactions.

"P2P3L6 - To me, 'declined' implies that this decrease is continuing - would 'decreased' here (and elsewhere) be clearer?"

Good point — in particular, the standard units for lightning stroke density (which are "year[-1]") make both "decline" and "decrease" sound like a rate of decline. Elsewhere, we generally say "declined by 40%", for which the units are less equivocal. We have amended P2P3L6 to say "lightning over the shipping lanes has decreased to an annual stroke frequency about 1 stroke km[-2]year[-1] lower than before the regulation".

"Fig 1 - A plot showing the current pattern might also be useful, as it is difficult to mentally subtract a linear scale form a log one."

We will add one to the supplemental that looks like this:

[Figure]

*Figure 2. Pre-IMO (top) and post-IMO (bottom) lightning stroke density.*

"P3P1L1 - Is assuming a linear ENSO effect sensible? There is significantly variation in the ENSO impact across regions."

The impact of ENSO can vary even within one region, and so an assumption of linearity is somewhat crude. However, intra-regional variability of ENSO impacts are accounted for by CAPE and Precip to some degree, with the ENSO index providing some information on large-scale, low frequency interannual variability. Indeed, the inclusion of the Niño Index improves the linear model slightly, adding around 10% explanatory power in the new regression.

"P3P1L3 - Are these variables really explanatory? There seems to be a large pattern of variation across the study region, does regressing by these factors remove it?"

See response to main concern.

"Fig 2 - The colours for the lines in the right hand subfigure are within the colorbar. Different colors (e.g. black, green) might be easier to read."

The figure has been amended to include a black line.

"Fig. 2 - As above, there appear to be a decrease in lightning far from the shipping lane in an approximately similar proportion (~25%). What is the reason for this?"

The improved regression indeed handles variations away from the shipping lane much better.

"P4P5L4(and elsewhere) - Nd misses a subscript."

Thank you, we have ensured that all references to $N_d$ are properly subscripted

"Fig 3 - Naming the rows in the figure would make this easier to read"

Agreed, we have added names to the rows.

"P6P2L1 - It is not clear to me that the Nd maintains previous levels in most regions. To the south of the shipping lane it appears there is a significant decrease in Nd post-IMO."

We make the statement "the $N_d$ away from the shipping lane *mostly* maintain their previous levels, as indicated by the overlap in the 95% confidence intervals (shading), particularly to the north". Here is a short explanation for that statement:

We assume the effect of the shipping lane may last several hours of downwind transport. With northerly winds averaging over 20km/hour in many NE monsoon months, we expect that the enhancement to extend up to 150km south. Beyond 150km south, the decreases are half insignificant, the other half being very narrowly significant. Meanwhile, 100% of the changes > 50km north of the shipping lane are insignificant consistent with northerly winds minimizing ship emissions being transported north of the shipping lane. Therefore, a majority of locations outside of the near-field downwind of the shipping lane are insignificant.

"P6P2L6 – Double ("

Thank you for your comment. We have removed the double parentheses

"P6P3L2 - A 40% decrease is relatively small, given the ~80% reduction in ship fuel sulphur content expected from IMO2020?"

We do not have an a priori expectation of how lighting stroke density should change in response to the fuel sulfur regulation. Quantifying and documenting that response is a motivation of this paper. Assuming the lightning enhancement is related to shipping induced CCN available for deep convective clouds, an 80% reduction in ship fuel sulfur content does not necessarily mean an 80% reduction in CCN, $N_d$ (McCoy et al, 2017), or the lightning enhancement. For example, from McCoy

et al (2017), $d\log_{10}(N_d)/d\log_{10}(SO_4)$ is expected to be on the order of 0.3-0.5, which is on par with an 80% change.

"P8P8L5 - Is such a high solar zenith angle common for MODIS retrievals in this region?"

No, but ruling out the effects documented by Grosvenor and Wood, ACP, 2014 is standard practice.

"PS2P5L4 - This is a very unusual referencing style, I would suggest the referencing in the SI should be the same as for the main manuscript."

Thank you for your comment. The referencing style has been fixed.

"PS2P6L1 - Why is reanalysis AOD used here? Previous studies (e.g. McCoy et al, JGR, 2017) suggest that reanalysis SO4 is a much better proxy."

Our point in showing the AOD data is to illustrate that the AOD products are not of much use directly. Some previous studies of lightning enhancements have utilized AOD as the aerosol(CCN) observable. In this region, there is no observable enhancement in AOD over the shipping lanes illustrating that the effects observed in $N_d$ and lightning are arising from radiatively inactive aerosol or otherwise aerosol(CCN) concentrations that are below the AOD detection capabilities. The lack of AOD enhancement is not surprising given that in the lightning season, cloud fractions are high, reducing the number of reliable AOD retrievals, and the sink to precipitation is strong. These aspects are motivation for the use of $N_d$ retrievals, which are more sensitive to aerosol number concentrations and more closely related to CCN at cloud base than AOD.

Furthermore, reanalysis $SO_4$ from MERRA-2 does not yet include the effects of the IMO regulation, such that ship emissions are still prescribed using pre-2020 emissions of $SO_2$ and $SO_4$.

"PS4P1L1 - Is the background AOD really low in this region? The MODIS Terra mean in this region is around 0.2-0.3. While this is not high, it is significantly above the MODIS noise floor and not what I would have classed as 'low'."

Agreed, the AOD is really not that low. We have amended the supplement to reflect this.

"Fig S2 - The SCS study region doesn't correctly cover the shiptrack in this diagram (or at least rendered with Acrobat Reader in Windows 11)."

I'm assuming that the region has rendered properly, but just in case, here is what we intended as Figure S2:

[Figure]

*Figure 3. A copy of supplemental figure 2, in the case that it's not rendering correctly for the reviewer.*

The SCS study region is the region to the right, for which the "shiptrack" box is the one to the northwest, and the "reference" region is the one to the southeast. These are the same regions used in Thornton et al (2017).

"Fig. S6 - While not significant at any particular distance, there does appear to be a consistent enhancement in the AOD post IMO to the north of the shiptrack. What is driving this and does it affect the Nd results presented earlier in this work?"

Merra-2 aerosol suggests that continental sources of black carbon and sulfate have been stronger in the post-IMO period. Taking a broader look at the region, here is the zonal mean across the whole Bay of Bengal (83-93E, 0-20N), where the shipping lane is at 6ºN:

[Figure]

*Figure 4. MERRA-2 black carbon mass zonal mean, averaged 83-93ºE.*

Sulfate shows a similar very small upward trend during the post-regulation period, while dust and sea salt show trends opposing the AOD trend. The increase in advection of biomass burning aerosol is already accounted for by thresholding corresponding Nd retrievals (see methods).

---

## Author Comment (AC2)

**Reviewer 1**

Specific comments:

"The abstract claims to test the sensitivity of lightning to "aerosol size distributions," yet the paper does not quantitatively present any results regarding aerosol sizes before and after IMO-2020. A better descriptor would be "aerosol concentration" or "aerosol emissions.""

We have updated the abstract to say "aerosol number-size distribution", to encapsulate the possibility that the production of viable CCN has been hindered by both reductions in total particle number and growth by sulfur oxidation and deposition.

"Figure 2 and associated discussion: does this analysis include both the Indian Ocean and South China Sea composited, or only the Indian Ocean? Please clearly state."
Thank you for your comment. We have updated the caption to clarify that it is a composite.

Page 3, 2D analysis of 3h CAPE / precip space:
1.  "What percent variance in lightning can be captured on the 3-hourly timescale, compared to the annual regression discussed earlier? Other works have indicated that CAPE and precipitation are not the best markers of convective strength over the ocean (see e.g. M.R. Igel 2014), so it would be beneficial to provide a quantification (and potentially brief discussion) of the predictive relevance of CAPE and precip in these data, rather than relying on Cheng 2021."

    With annually averaged observations, the assumption of linear relationships between CAPE, precipitation, and lightning over the ocean are more reasonable. With 3-hourly data, we see clearly non-linear relationships between these three variables, and therefore choose to use the CAPE-precip space to assess differences. Similar to Cheng et al (2021), we observe threshold-like behavior:

[Figure]

*Figure 1. Lightning stroke frequency in CAPE-Precip space, pre-IMO regulation in the Bay of Bengal.*

    Where low-CAPE and low-precip environments likely lack the necessary updraft strength or vertical moisture flux to frequently generate charge separation.

If we repeat the analysis in Cheng et al (2021), using the single 3-hourly sqrt(CAPE) threshold of 15 m/s, we see that greater than 35-55% of spatial variability in lightning stroke density is explained by CAPE*Precip across the two oceanic regions of interest. As shown in the figure above, the advantage of the CAPE-precip space is that it does not require an *a priori* assumption about the functional relationship between CAPE, Precip, and lightning, nor of a specific threshold behavior. We have added the above figure to the supplement.

2. If following Cheng 2021, Fig. 2 should exclude points where CAPE^(1/2) < 15 m/s.
   Agreed. We have removed 3-hourly observations where CAPE^(1/2) < 15 m/s from consideration in the regression for Figure 2.

3. It is potentially interesting that lower-CAPE retrievals show both a stronger pre-IMO enhancement, and a greater difference following IMO, particularly in the South China Sea. This would indicate that weaker systems (lower CAPE) are more susceptible to aerosol, which may warrant a brief discussion.

   Thank you for your comment. We agree and have added a brief note to this effect.

"Page 4, discussion of optical thickness, reads "We have partially accounted for….using MERRA-2 reanalysis estimates…in constructing Figure 4." However, it is not clear from the figure caption how this correction is performed. Presumably the phrases at the end of the Appendix explain this correction, and should be referenced in the text accordingly."

We have added the explanation to the figure caption

"The Supporting Information would benefit from subheadings to organize and divide contents. References to the SI in the main text would then be more precise."

Thank you for the comment. We have added subheadings for organization

SI figure 3:
   1. "Clarify whether the data displayed are 3-hourly or annual mean"
      Thank you for the comment. We have clarified that these are annual means
   2. "SI page 2 indicates that S3 shows data "outside of the shipping lane", but the figure appears to include all data, including over top of the shipping lane."
      We have clarified the language to indicate that data both outside and over top the shipping lane are included

"ACP Data Availability policy requires that data which cannot be deposited publicly because of commercial constraints should include a detailed explanation of why this is the case, and additionally that the data should be made available to reviewers. The existing statement in the manuscript only directs the reader to WWLLN.net, where data are only accessible for a fee, and should be updated to reflect ACP's requirements."

WWLLN lightning location data are collected by a global scientific collaboration and managed by the University of Washington.  The WWLLN collaboration receives no federal, state or private funds to pay for the network operations, which are fully paid for by data sales. The University holds a copyright on the dataset to protect the redistribution of the data by unauthorized persons.  Therefore, the stroke-level data is not free to the public. The composited annual stroke densities (as a function of distance from the shipping lane) and the mean pre- and post-regulation stroke densities region-wide are provided as part of the Zenodo code supplement. We have clarified this in the manuscript. If the editors would like to check the results of this paper by looking at the stroke level data, that can be arranged, if the editors will sign a nondisclosure agreement.

"I suggest the authors confirm that the manuscript falls within ACP's 2500 word limit – a cursory word count on my part read 2700, but this included in-text citations which may not count toward the limit."

Thank you for the comment. We have removed 200 words to bring the manuscript under the word limit. Perhaps the editor can confirm whether we were, in fact, over the limit and whether we are now under it.